# The Association between ADHD and Celiac Disease in Children

**DOI:** 10.3390/children9060781

**Published:** 2022-05-25

**Authors:** Sonia Gaur

**Affiliations:** Department of Psychiatry, Stanford School of Medicine, 401 Quarry Road, Stanford, CA 94305, USA; gaurs@stanford.edu; Tel.: +1-(650)-567-5689

**Keywords:** ADHD, celiac disease, systematic review, association

## Abstract

Controversy around the association between celiac disease (CeD) and attention deficit hyperactive disorder (ADHD) was addressed by a systematic review in 2015, ultimately showing no association. Since 2015, there have been several studies showing an association between celiac disease and attention deficit hyperactive disorder. This is an updated systematic review. Background: Most experts agree on the recommendation to not screen as part of the standard of care for ADHD in persons with CeD or vice versa. Simultaneously, they propose that untreated patients with CeD and neurological symptoms such as chronic fatigue, inattention, pain, and headache could be predisposed to ADHD-like behavior, namely inattention (which may be alleviated by following a gluten-free diet). The inattentive subtype of ADHD that encompasses the symptoms of inattention is phenotypically heterogeneous, as it includes the clinical construct of sluggish cognitive tempo (SCT). SCT symptoms overlap with the neurological manifestations of CeD. Methods: A systematic search (PRISMA) of PubMed, Google Scholar, EMBASE, Web of Science, Stanford Lane, SCOPUS, and Ovid was conducted for articles up to 21 February 2022. Of these, 23 studies met the criteria. Results: Out of the 23 studies, 13 showed a positive association between ADHD and CeD. Most studies that showed a positive association had been published in the last five years. Inconsistencies in the results remain due to the heterogeneous methodology used, specifically for ADHD and the outcome questionnaires, as well as a lack of reporting on ADHD subtypes. Conclusion: There is an association between ADHD and celiac disease. The current methodological limitations will be lessened if we examine the subtypes of ADHD.

## 1. Introduction

The prevalence of celiac disease is estimated to be 0.5–1.4% worldwide [1], with the rate of psychiatric illness in patients with untreated celiac disease (CeD) being as high as 21% [2]. In children with CeD, the risk of developing neuropsychiatric disturbances is only 2.6%, compared to 26% in adults [3]. The mechanisms involved in the origin and pathogenesis of the mental and behavioral disorders related to celiac disease in adolescents is unknown, but it has been suggested that the ingestion and cleavage of gluten into immunogenic peptides could lead to the leaking of peptides through the intestinal wall. These peptides may then traverse the blood–brain barrier and potentially induce low-grade inflammation in the brain [2]. These immunological mechanisms may contribute to ADHD development and manifestation [4,5,6]. Functional deficits include the hypoperfusion of cerebral regions, primarily in the frontal cortex, in untreated adult patients with CeD, but not in treated patients [7,8]. This under-activation in the right ventrolateral prefrontal cortex is seen in ADHD [9]. In a meta-analysis, researchers detected a significant increase in neurodevelopment conditions, namely attention deficit hyperactivity disorder (ADHD) (OR, 1.39; 95%CI, 1.18–1.63; *p* < 0.0001), among the CeD population compared to healthy controls but no significant differences for bipolar disorder or schizophrenia [10]. Meanwhile, adults with ADHD showed an increased occurrence of obesity, migraines, sleep disorders, asthma, and CeD [11].

CeD can occur in healthy persons (silent CeD) or later in life in persons with positive gluten-sensitive autoantibodies but a normal to minimally abnormal mucosa upon intestinal biopsy (potential CD) [12]. This period prior to the diagnoses of CeD is when ADHD symptomatology is high in frequency and severity and could be treatment-refractory without a gluten-free diet (GFD) [13,14,15], which is why this association is frequently studied.

ADHD is a neurodevelopment condition diagnosed by the International Classification of Disease (ICD) criteria. Depending on the study year, professionals have used either ICD-9 or ICD-10. The difference between years of study is relevant when we consider the subtypes of ADHD. In the ICD-10, the ADHD subtypes have separate codes, namely, inattentive (F90.0), hyperactive (F90.1), combined (F90.2), other (F90.8), and unspecified (F90.9)—whereas the ICD-9 has only three codes: combined (314.1), inattentive (314.00), and residual (314.8). These limitations have been discussed in previous studies [16,17]. Although the dimensions of ADHD are recognized as valid and clinically meaningful, some researchers have found that they are amenable to better specification, in particular, the inattentive type (IN), which can be phenotypically heterogeneous as it encompasses the sluggish cognitive tempo (SCT) subtype [18,19,20,21].

The symptoms associated with SCT are characterized by a pattern of inconsistent alertness and orientation, confusion, physical under-activity, daydreaming, and lack of mental alertness. Prior studies have found SCT to be associated with poor/abnormal neurocognitive performance, even when controlling for other ADHD symptoms [22,23]. Given this heterogeneity, it is imperative that we identify this population, which can contribute to interpreting the confounding results seen in studies of the association. Given that CeD is associated with a low quality of life (HRQoL scores) for both children and their caregivers [24], and ADHD increases the risk of depression [25] later in life, this review can help in decreasing the factors that contribute to depression.

A past systematic review of eight studies on the association of ADHD and CeD concluded that there was no association [26]. Overall, there have been 23 studies examining the association, and this represents an update of their results. To the best of our knowledge, there have been no studies that have examined the subtypes of ADHD in relation to CeD.

## 2. Materials and Methods

This review was conducted following the PRISMA reporting guidelines (Figure 1).

### 2.1. Search Strategy

We conducted a literature search (see Appendix A/Table A1 for the list of search terms). Our search strategy was verified by checking it against an independent search created by a librarian at Stanford’s medical library. Articles published before 21 February 2022 were included in the search. Lower date restrictions were not used, except for Google Scholar (2015–2022). The electronic databases PubMed, Google Scholar, EMBASE, Web of Science, Stanford Lane, SCOPUS, and Ovid were searched for relevant journal articles. This review protocol was registered in the International Prospective Register of Systematic Reviews (PROSPERO).

The inclusion criteria were as follows: (1) search terms of ADHD and CeD (noncontrolled and controlled trials/database/registry); (2) ages less than 18 years (including three studies with a range of ages and reported data that included subjects below 18 years). In addition, we considered the past study’s limitations/strengths and the reference list of the included studies to find other relevant articles. Only articles in English were reviewed. Posters from conferences were reviewed, and attempts were made to access unpublished manuscripts, but the authors did not have the data ready. There were no ineligible studies.

The exclusion criteria were: (1) studies that included the above search terms but did not narrow the diet intervention to gluten only; (2) studies that included other related diagnoses (e.g., inflammatory bowel disease/IBD, Crohn’s, and autism spectrum disorders) and did not report the results that included the search terms; and (3) review articles that did not contain data that established an association between ADHD and CeD.

### 2.2. Screening and Selection of Studies

A total of 52 titles and abstracts were retrieved and screened. Full-text articles were read to evaluate the inclusion or exclusion criteria for purposes of association. In all, 23 met both the inclusion and exclusion criteria. Out of these 23 studies, 8 had been reported in the previous review [16].

### 2.3. Outcome Measures

The outcome measures were the reported association of the two conditions. Data extraction included: (1) the mean age and range; (2) the assessment process to validate CeD and ADHD; (3) the study design; (4) the sample size; and (5) the reported association.

## 3. Results

Table 1 shows the association between ADHD and CeD. A total of 23 studies from the 52 screened articles were included. Out of the 23 studies, there were 13 studies that showed an association. Most of the studies that showed a positive association were published after 2015. Those studies that showed no association were reviewed to understand their reported limitations. The limitations of studies are described below as a narrative owing to the table format. It was found when a structured psychiatric assessment in children (KSADS) was used retrospectively, it affected the validity of the study [15]; another diagnosed ADHD upon a neurological examination without mentioning the ICD criteria [3]. Certain studies used a small number of subjects. For example, Lichtwark et al. [27] used 11 subjects, although this may be a small number, given the authors’ extensive cognitive testing, the results are meaningful and were not considered a limitation; moreover, the authors did not identify it as a limitation. An earlier study did not use all of the criteria to diagnose CeD [28]. The largest population based study to address psychiatric conditions in CeD highlighted that ADHD showed an association of OR = 1.75 in 112,240 patients with CeD [29].

The studies diagnosed ADHD using DSM-IV (*n* = 12), DSM-5 (*n* = 1), ICD-9 (*n* = 5), and ICD-10 (*n* = 6), and some (*n* = 3) did not report any criteria. The method of assessment of ADHD and its symptomatology varied and ranged from using standardized methods to historical data or a neurological/psychiatric examination (see Table 1). Diagnosis of CeD or its symptomatology varied in terms of methodology, dependent on the study aim. Most study data were of children and adolescents, but some studies included adults with a mean age of less than 22 years. A number of studies related to the association between ADHD and CeD approached it using subjects with CeD with a GFD as an intervention (*n* = 6) or examining the ADHD (*n* = 8) and CeD (*n* = 9) population via medical database/registry/chart reviews. All of the studies used serological markers for CeD when ADHD was the target population.

The reviewed results indicate that ADHD symptom severity correlates in patients with CeD compared to the general population, and a six-month gluten-free diet (GFD) improved ADHD symptoms in most patients [30,31]. Neurological symptoms were found to correlate with Marsh type 3B pathology [36] versus the non-neurological presentation of CeD. In newly diagnosed adults with CD, a significant improvement in cognitive functioning when examining “brain fog” with CeD using the Subtle Cognitive Impairment test (SCIT), particularly verbal fluency, attention, and motor function, was noted after a 12-month adherence to GFD [27]. The TRACE study, a two-armed randomized controlled trial comparing the short- and long-term effects of an elimination diet and a healthy diet that differentiates the gluten-free diet in children with ADHD, is yet to be published [46].

The studies did not report subtype differences in the association between CeD with ADHD, a majority did not look at subtypes. In examining the evidence of dietary interventions in the subtypes of ADHD in the autistic population, the hyperactive (HYP) subtype showed a downward decline with a sustained decrease that lasted up to 12 months, while the IN subtype showed an improvement at 8 and 12 months [47,48,49]. Comparable results between individual scores of IN and HYP in the ADHD population using the ADHD-IV checklist were seen [40]. Prior to treatment, patients with CeD had significantly higher total ASRS scores, inattention sub-scores, and hyperactivity sub-scores than healthy controls, showing a larger improvement in the inattention versus hyperactivity domain of 4.4 versus 1.9 [41].

## 4. Discussion

This systematic review indicated an association between ADHD and CeD. Ongoing methodological limitations have contributed to meaningful recommendations. Although the criteria used for diagnoses of CeD were closely followed, diagnosis of ADHD using ICD criteria was identified as a limitation, as it may have allowed for the inclusion of misdiagnoses and diagnoses [17,18]. This limitation may be explained by one of the subtypes of ADHD, called SCT.

In terms of diagnostic validity, there is currently not enough evidence to describe the subtype of SCT in diagnostic terms due to poor negative predictive power toward an ADHD diagnosis [50]. Although SCT items and DSM-IV inattention symptoms are highly correlated, a subset of SCT items does not load on either of the inattention or hyperactivity factors. The psychometric validity of the SCT construct is seen in children and adolescents [23,24]. A pattern of inconsistent alertness and orientation, confusion, physical under-activity, daydreaming, and lack of mental alertness characterize the SCT factor [20]. These appear to be more in the realm of the extra-intestinal symptoms of CeD. If we subtype ADHD to include SCT in individuals with CeD it may help to address the methodological limitations in future studies. This may help us identify a behavioral phenotype of ADHD in a high-risk population of celiac disease

This review supported previous evidence where ADHD symptoms were present prior to the diagnoses of CeD, as seen in atypical CeD [14]. In a population-based study with a median follow-up period of 12.3 years, psychiatric disorders were also more common before a diagnosis of celiac disease (odds ratio, 1.56; 95 CI, 1.39–1.76) with attention deficit hyperactivity disorder (HR, 1.29; 95%CI, 1.20–1.39) [13]. Specifically, children with newly diagnosed or potential CeD often complained of aches and pains, tired easily, were easily distracted, and had trouble concentrating [4].

Since the prevalence of celiac serology is higher in ADHD children than in the general population [51], all of the studies used serological markers for CeD when ADHD was the target population. However, the studies were mixed on this aspect as well. The prescription of diets based on IgG blood tests has not been encouraged [40], but in a study of patients with suspected ADHD, some scientists recommended screening for this disease via tissue anti-TG IgA antibody determination during normal practice [31]. Given the fact that extra-intestinal manifestations dominate the clinical presentation of over half of patients, the combination of a careful case-finding strategy and a more liberal use of serological tools to improve the detection rate of CeD is warranted [52].

For more complex genetic connections, the predisposing factor for CeD is the HLA-DQ2 and -DQ8 haplotypes [53,54,55]. This HLA complex genetic system encodes proteins that regulate immune/inflammatory processes and has been shown to be associated with psychiatric disorders [56]. As a result of the shared genetic background, CeD has been found to be significantly associated with ADHD in first-degree relatives (FDRs) [55], along with a positive co-relation with other autoimmune conditions [56]. In a small sample of ADHD patients, 30–40% of the population had susceptibility alleles to celiac disease DQ2 and 33% had the DQ2 allele [16]. A Danish national register study of almost a million people found that children with autoimmune disease were 24% more likely to develop ADHD. Maternal but not paternal autoimmune disease was associated with a 12% greater likelihood of ADHD in their offspring [57,58]. In contrast, an association between childhood ADHD and immune-mediated diseases, such as type I diabetes, juvenile rheumatoid arthritis, and asthma, was found, but no association was recorded between IBD and celiac disease [59].

It has been shown that environmental (as opposed to genetic) factors are more relevant for the SCT type versus the HYP and IN subtypes of ADHD. This was seen in a twin study, where individual differences in the SCT subtype were mostly explained by non-shared environmental factors, while genetic factors mostly influenced the variability in the HYP and IN subtypes. This finding indicates that, when we disregard SCT, it may cause greater ADHD heterogeneity and diminished diagnostic clarity among children belonging to the ADHD inattentive group [60].

The prevalence of celiac disease varies with sex, age, and location [61]. Gender-related transmission was highlighted when a global examination of temporal trends showed that the incidence of CeD is highest in females and in children [62]. When examining first degree relatives (FDRs) in a large meta-analysis, the pooled prevalence of CeD among FDRs of patients with CeD was 7.5%, with sisters and daughters of the index patient with CeD at the highest risk, which varied with geographical location [1]. When adjusting for socioeconomic status, region of residence, and family history of psychiatric disorders, the association between ADHD and maternal celiac disease was only significant in daughters [20]. However, one cannot rule out that the cognitive score in middle-aged women without celiac disease is not significantly associated with a GFD [63]. This emphasizes the importance of targeting the phenotype associated with cognitive impairments.

The limitations of this review are: (1) there was no comparison of the articles used for the review to minimize bias as there was one author, and one could infer unconscious bias; (2) the SCT is a well-known subtype of ADHD, but it cannot be measured if we use the ICD criteria, and most of the standardized questionnaires are standardized in a white population [20]; (3) the number of studies was small and differed in study design, population screened, interventions, controlled versus noncontrolled; (4) the race/ethnicity of the participants was not reported and could not be used in a meaningful manner; thus, it was not clear whether these samples were representative of race/culture and the obtained results generalized to more diverse populations; and (5) the two pilot studies [16,27] included in the table raised concerns about the risk of generalizability biases; this may not preclude them from the review, but it is recognized as a limitation if future studies are to be designed [64].

## 5. Conclusions

There was an association found between ADHD and celiac disease, which contrasts with the previous review. This review highlighted the need to examine the subtypes of ADHD, specifically the inattentive type. There may be a behavioral phenotype of ADHD that responds to a gluten-free diet. This is consistent with earlier studies and supports the screening of ADHD patients for celiac disease. It also supported the previous review’s recommendation for clinicians to assess a broad range of physical symptoms in addition to typical neuropsychiatric symptoms when evaluating patients with ADHD.

## Figures and Tables

**Figure 1 children-09-00781-f001:**
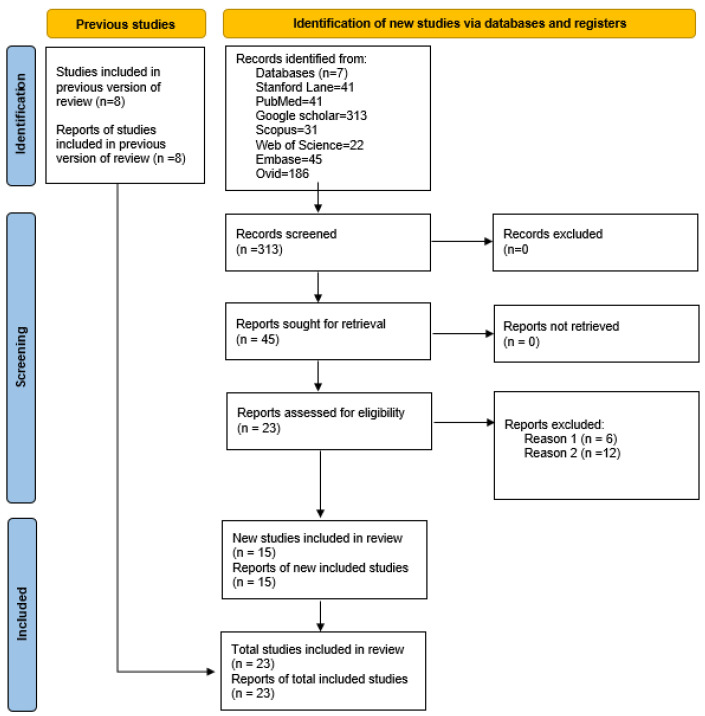
PRISMA 2020 guidelines for updated systematic reviews (http://www.prisma-statement.org/, accessed on 21 February 2022).

**Table 1 children-09-00781-t001:** Association between ADHD and celiac disease.

Source	Study Design	Inclusion Criteria	No. of Subjects	No. of Controls	Age Range ± Mean (years)	ADHD Criteria	ADHD Method of Diagnosis	Celiac Disease (CeD) Serology/Biopsy/HLA	Association
Lahat, 2000 [28]	OB, CS	ADHD	39	34	6–13, 8.6	DSM-IV	Nil	IgA, IgG, AGA, EMA	No
Zelnick, 2004 [30]	OB, CO, CC	CeD/GFD	111	211	20.8 ± 8.9	DSM-IV	Neurological examination	IgA, EMA, biopsy	Yes
Pynnonen, 2004 [15]	OB, CO, RET	CeD/GFD	29	29	12–17	DSM-IV	KSADS-PL, CBCL, YSR	EMA, tTGAb, biopsy	No
Niederhofer 2006 [31]	OB, COH, RET	CeD	132	0	3–57, 19.3	ICD-10, DSM-IV	Hypescheme	EMA, tTGAb	Yes
Ruggieri, 2008 [3]	OB, CO, COH, PRO	CeD	835	0	9 mths–17, 7.3	Nil	Neurological examination	AGA, EMA, HLA, biopsy	No
Niederhofer2011 [32]	OB, COH, PRO	ADHD	67	0	7–42, 11.4	ICD-10, DSM-IV	Hypescheme	EMA, tTGAb	Yes
Dazy, 2013 [33]	OB, CO, CC	CeD	281	301	<25	ICD-9	Nil	EMA/tTG, biopsy	No
Diaconu, 2013 [34]	PRO	CeD/GFD	48	0	2–18	DSM-IV	Psychological examination	Biopsy	No
Gungor, 2013 [35]	OB, CO, CC	ADHD	362	390	5–15	DSM-IV TR	KSADS	tTgIgG, tTgIgA, biopsy	No
Lichtwark, 2014 [27]	LO, PS	CeD/GFD	11	0	22–39, 22	Nil	SCIT, Trails	Nil	Yes
Isikay, 2015 [36]	CS, PRO	CeD	40	297	<18	Nil	Neurological examination	tTG, biopsy	Yes
Kumpersack, 2006 [37]	PRO	ADHD	53	0	13.2	ICD-10	Nil	ESPGHAN	No
Butwicka, 2017 [38]	COH, POP, PRO	CeD	10,903	12,710	<18	ICD-9, ICD-10	Adult ADHD Self-Report Scale	Nil	Yes
Prinzbach, 2018 [39]	COH, RET	CeD	433	4330	0–21, 9.53	ICD-10	Nil	Nil	No
Canal, 2019 [16]	PS	ADHD	6	0	9–13	DSM-5	CPT-III	ESPGHAN	Yes
Honar, 2019 [40]	CS	ADHD	99	0	4–18	DSM-IV	ADHD checklist	tTGAb	Yes
Kristenson, 2019 [41]	LO, PRO	CeD	31	60	>18	DSM-IV	Adult ADHD Self-Report Scale	Nil	Yes
Lee, 2020 [42]	PRO	CeD/GFD	33	0	6–18,10.7	DSM-IV	CPT-III, CBCL	tTIg/IgG	Yes
Coburn, 2020 [43]	PRO	CeD/GFD	73	0	3–18	DSM-IV	Mental health diagnosis	Nil	Yes
Kedem, 2020 [17]	RET, COH	ADHD	117	0	17–35	ICD-9	Nil	ESPGHAN	No
Kirsacliglu, 2020 [44]	PRO	ADHD	117	0	6–18	DSM-IV	Nil	ESPGHAN	No
Lebwohl, 2021 [13]	COH	CeD	3174	13,286	<18	ICD-9, ICD-10	Nil	Nil	Yes
Alkhayyat, 2021 [29]	OB, COH, RET	CeD	OR=1.75	112,240	All age ranges	ICD-9	Nil	Nil	Yes

OB, observational; CS, cross-sectional; CO, comparative; CC, case–control; LO, longitudinal; PS, pilot study; PRO, prospective; RET, retrospective; COH, cohort; POP, population; CeD/GFD, CeD on a GFD; AGA, antigliadin antibody; EMA, anti-endomyosial antibody; tTGAb, anti-tissue transglutaminase antibody; tTG, tissue transglutaminase; NCGS, anti-gliadin IgA/IgG antibodies; IgG, immunoglobulin G; IgA, immunoglobulin A; HLA, human leucocyte antigen DQ2/DQ8; tTgIgG/IgA, anti-tissue transglutaminase IgA/IgG; ESPGHAN [45], European Society for Pediatric Gastroenterology Hepatology and Nutrition; CPT-III, Conners Continuous Performance Test III; YSR, Youth Self-Report; CBCL, Child Behavior Checklist; DSM IVTR/5, Diagnostic and Statistical Manual of Mental disorders IV/5; ICD- 9/10, International Classification of Disease 9/10; SCIT, Subtle Cognitive Impairment Test, OR, odds ratio of ADHD.

## Data Availability

Not applicable.

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
