# Peer review of "The Association between ADHD and Celiac Disease in Children"

_children, 2022, doi:10.3390/children9060781_

Round 1

Reviewer 1 Report

Thank you for letting us review your manuscript. My recommendations are as follows:

  1. Overall needs grammar revisions
  2. Please use third person sentences vs. First person
  3. All the Abbreviations should be addressed in the beginning vs. in discussion such as GFD 
  4. Figure 1.0 needs major revision
    1. Can Author state the brief reasons for the exclusion of such studies ( from 313 to 23 studies) 

      Is it possible to remove the symbols ⥥ from Figure 1

      Can author elaborate what is previous version of review with n=8 ?

      Figure 1 identification to inclusion ( left panel) numbers do not match with text.

  5. Table 1.0 - Needs major revisions
    1. Table 1.0 Does the author mean Hypescheme instead of HYPERSCHEME ?

      Table 1.0 Is there a study missing here, where it says (64)

      Table 1.0 needs to be rearranged so that criterias are seen clearly respective to their studies

      Can the author add limitation/remarks to each study listed whether they showed association or not e.g. Lichtwark 2014 has only 11 patients. 

      Coburn 2020 ; diagnosis of ADHD is solely based on parent questionnaire which may not be ideal for this particular review

      Table 1.0 Lebwohl 2021 does state the method if identification of celiac disease as Biopsy, and ADHd via ICD, And please double check case/control numbers. please update the findings. 

      Study reference 56 in table 1.0 please double check the case/control numbers per study

  6. Limitation : there is inclusion of two pilot studies in this review that will skew results ( bias)
  7. Conclusion: results are encouraging, i would recommend to decrease the tone for strength of association here and avoid from drawing strong conclusion as studies are really heterogenous ( as described in limitations)

Author Response

Gratitude for this reviewer's comments. The points made are excellent and authors have grappled with some of these thoughts too. Author has written response to the reviewer. We cannot address all of them as I am waiting for editing services. It is my hope that the editing services can help with flow and clarity.

Overall needs grammar revisions 

I have asked the editing services to improve upon the manuscript. It si in process. 

 1) Please use third person sentences vs. First person 

Yes, I agree that it can be confusing to the reader. The first-person usage is in the Method section. I have changed it to be the 3rd person. 

2)All the Abbreviations should be addressed in the beginning vs. in discussion such as GFD  

I appreciate this lapse on my part. I have addressed it. 

3)Figure 1.0 needs major revision 

-Can Author state the brief reasons for the exclusion of such studies ( from 313 to 23 studies) 

-Is it possible to remove the symbols ⥥ from Figure 1

-Can author elaborate what is previous version of review with n=8 ?

-Figure 1 identification to inclusion ( left panel) numbers do not match with text. 

The author is indebted to the reviewer as the numbers had to be changed. You notice that additional help for my search had to be undertaken and my narrative reflects the changes: Example new accurate no's in the figure will be added. It has not taken away from the same results, and I addressed the discrepancies. I would acknowledge that the reviewer has in-depth knowledge of the field, and the author is grateful that they agreed to this review. 

4)Table 1.0 - Needs major revisions 

-Table 1.0 Does the author mean Hypescheme instead of HYPERSCHEME ? 

Yes you are correct ! 

-Table 1.0 Is there a study missing here, where it says (64) 

It is a formatting issue, and the information is in the line of Niederhofer but shows it is below the name. 

-Table 1.0 needs to be rearranged so that criterias are seen clearly respective to their studies 

I agree. It will be formatted. 

-Can the author add limitation/remarks to each study listed whether they showed association or not e.g. Lichtwark 2014 has only 11 patients. 

This is a four-part concern: 

1)The Table does show the association as a yes or no. It may not be apparent due to formatting. 

2)The limitation/remarks are in the discussion. I have addressed collectively them under their relevance.  

3) The limitation of each study cannot be addressed in the tabular format. Attempting to do this may result in a separate table and may not improve upon the discussion. Also, this is an updated review, explanation of limitations may confuse the reader because the tabular form will cause me to shorten the explanation. This brevity may confuse the reader. Having said that one can consider it but will need more time.

4)I have added: 

“There are certain studies that used a small number of subjects, for example Lichtwark et al [36] used 11 subjects, this may be a small number but given the authors extensive cognitive testing the results are meaningful and given the context, I would agree with the study authors that it is, not a limitation.” 

-Coburn 2020 ; diagnosis of ADHD is solely based on parent questionnaire which may not be ideal for this particular review 

Yes the way the author has reported it is misleading. I will correct the table to change from Parent Report to Mental Health diagnosis as reported in Coburn et al [59]. This will help reviewer or reader understand the relevant report. This may help explain the relevance of the study because the subjects were diagnosed by a mental health professional versus in other studies by neurologists, questionaries, etc supporting the finding that when the diagnosis of ADHD is specific the studies show an association between CeD and ADHD” 

-Table 1.0 Lebwohl 2021 does state the method if identification of celiac disease as Biopsy, and ADHd via ICD, And please double check case/control numbers. please update the findings 

The formatting is creating this miscommunication,  it is ICD 9 and 10. This will be addressed by formatting. 

. 
-Study reference 56 in table 1.0 please double check the case/control numbers per study 

I agree they contrast with the other studies, but in the discussion, they have been referenced. It is checked and this is what they report.  

5)Limitation : there is inclusion of two pilot studies in this review that will skew results ( bias) 

Yes I agree, one has to be specific with the bias, I will add the statement and reference. 

 “The two pilot studies [17,36] included in this table bring concerns of risk of generalizability biases, this may not preclude them from the review but are recognized as a limitation if future studies are to be designed [66].Beets, M.W., Weaver, R.G., Ioannidis, J.P.A. et al. Identification and evaluation of risk of generalizability biases in pilot versus efficacy/effectiveness trials: a systematic review and meta-analysis. Int J Behav Nutr Phys Act 17, 19 (2020). https://doi.org/10.1186/s12966-020-0918-y)” 

6)Conclusion: results are encouraging, i would recommend to decrease the tone for strength of association here and avoid from drawing strong conclusion as studies are really heterogenous ( as described in limitations) 

Response: 

“There is an association between ADHD and Celiac disease, contrasting with previous review. Limitations regarding heterogeneity in the studies continue and recent studies have tried to address them. Perhaps if we examine the subtypes of ADHD, specifically the inattentive type, we may decrease the limitations. There is evidence to show that a behavioral phenotype of ADHD can respond to a gluten-free diet. It is consistent with earlier studies and supports screening ADHD for Celiac disease. It supports the previous reviews recommendation for clinicians to assess a broad range of physical symptoms, besides typical neuropsychiatric symptoms, when evaluating patients with ADHD.” 

Reviewer 2 Report

In this review article, the Authors aimed to address the association between Attention Deficit Hyperactive Disorder (ADHD) and Celiac disease (CD). They performed a systematic search of PubMed, Google Scholar, EMBASE, Web of Science, Stanford Lane, SCOPUS, for articles up to February 2022.

Out of twenty-three studies selected, thirteen showed a positive association between ADHD and CD. Most studies that showed a positive association was in the last 5 years. The review showed that the criteria for CD have been followed to address past limitations, but the reporting of ADHD continues to be inconsistent, specifically in the outcome questionnaires as well as lack of reporting of ADHD subtypes. They concluded that there is an association between ADHD and Celiac disease. The methodological limitations continue in studies and will be improved if the subtypes of ADHD are considered.

The study is of interest and the topic is of clinical relevance. To further support the rationale of this clinically relevant association, the Authors should however further recall literature data demonstrating an association between untreated celiac disease and neurologic disorders that are characterized by the serological positivity of antineuronal antibodies (Sera of patients with celiac disease and neurologic disorders evoke a mitochondrial-dependent apoptosis in vitro. Gastroenterology. 2007 Jul;133(1):195-206) and antiganglioside antibodies (Anti-ganglioside antibodies in coeliac disease with neurological disorders. Dig Liver Dis. 2006 Mar;38(3):183-7; Anti-ganglioside antibodies and celiac disease. Allergy Asthma Clin Immunol. 2021 May 28;17(1):53. ) as previously reported.

 -To further improve the clinical impact of the study, in the discussion, it would be also discussed the other extraintestinal manifestation of untreated celiac disease characterizing children such as atopy as previously demonstrated (Prevalence of silent coeliac disease in atopics. Dig Liver Dis. 2000 Dec;32(9):775-9) to further highlight that celiac disease can be suspected also in patients with atypical presentation.

Author Response

Dear Reviewer,

Thank you for your suggestions to improve upon this review. I understand you have asked for two responses or made two suggestions. Apologise if I missed any other feedback.

In this review article, the Authors aimed to address the association between Attention Deficit Hyperactive Disorder (ADHD) and Celiac disease (CD). They performed a systematic search of PubMed, Google Scholar, EMBASE, Web of Science, Stanford Lane, SCOPUS, for articles up to February 2022. 

Out of twenty-three studies selected, thirteen showed a positive association between ADHD and CD. Most studies that showed a positive association was in the last 5 years. The review showed that the criteria for CD have been followed to address past limitations, but the reporting of ADHD continues to be inconsistent, specifically in the outcome questionnaires as well as lack of reporting of ADHD subtypes. They concluded that there is an association between ADHD and Celiac disease. The methodological limitations continue in studies and will be improved if the subtypes of ADHD are considered. 

The study is of interest and the topic is of clinical relevance. To further support the rationale of this clinically relevant association, the Authors should however further recall literature data demonstrating an association between untreated celiac disease and neurologic disorders that are characterized by the serological positivity of antineuronal antibodies (Sera of patients with celiac disease and neurologic disorders evoke a mitochondrial-dependent apoptosis in vitro. Gastroenterology. 2007 Jul;133(1):195-206) and antiganglioside antibodies (Anti-ganglioside antibodies in coeliac disease with neurological disorders. Dig Liver Dis. 2006 Mar;38(3):183-7; Anti-ganglioside antibodies and celiac disease. Allergy Asthma Clin Immunol. 2021 May 28;17(1):53. ) as previously reported. 

Response: 

The above referenced article is an important study proposing a new marker, namely anti-ganglioside antibodies. The article broadly mentions the connection between CeD and neurological disorders, naming several conditions except ADHD. Authors conclude that introducing this article or marker can be done in future articles or even in a clinical trial, once the association of CeD and ADHD has fewer existing heterogenous variables. Authors thank reviewers for directing them to this potential marker. 

-To further improve the clinical impact of the study, in the discussion, it would be also discussed the other extraintestinal manifestation of untreated celiac disease characterizing children such as atopy as previously demonstrated (Prevalence of silent coeliac disease in atopics. Dig Liver Dis. 2000 Dec;32(9):775-9) to further highlight that celiac disease can be suspected also in patients with atypical presentation. 

Response: 

Authors thank reviewers for directing them to the article of CeD and atopy. Atopy as a broader term is seen with ADHD but this referenced article does not mention ADHD and atopy anywhere in the article. That may appear confusing to the reader and would add another reference to an extensive list of references already present. Given it is a topic not covered in this review, the authors conclude that introducing this article does not add to the discussion and may be addressed as atopy and ADHD in future reviews/studies.